# Risk and Influence Factors of Fall in Immobilization Period after Arthroscopic Interventions

**DOI:** 10.3390/jpm12111912

**Published:** 2022-11-16

**Authors:** Johannes Rüther, Luka Boban, Christoph Paus, Kim Loose, Maximilian Willauschus, Hermann Josef Bail, Michael Millrose

**Affiliations:** 1Department of Orthopedics and Traumatology, Paracelsus Medical University, Breslauer Straße 201, 90471 Nuremberg, Germany; 2Department of Trauma and Orthopedic Surgery, Schwarzwald-Baar Clinic, 78052 Villingen Schwenningen, Germany; 3Department of Trauma Surgery and Sports Medicine, Garmisch-Partenkirchen Medical Centre, 82467 Garmisch-Partenkirchen, Germany

**Keywords:** knee surgery, arthroscopy, sports injury, orthosis, postoperative treatment

## Abstract

Knee injuries are one of the most common injuries. Falls during the immobilization period can deteriorate the postoperative outcome. The risk factors causing falls after initial injury and the question of whether a rigid orthosis serves as a protective factor remain unclear. The primary aim of the study was to record the fall rate in the first six weeks after arthroscopic intervention. The secondary aim was to assess the influences of risk factors and protective factors on these fall ratios. Different scores were examined and compared in the groups ‘fall event’ and ‘no fall’. Data from 51 patients (39 males, 12 females) with a mean age of 31.2 years (19–57 years) were collected. A total of 20 patients suffered at least one fall event within the observation period. A total of 18 of 23 fall events happened within the first three weeks postoperatively. The Extra Short Musculoskeletal Function Assessment Questionnaire (XSMFA) showed a significant difference between the groups (*p* = 0.02). People with multiple injuries to the knee joint were more likely to suffer fall events. Conclusively, patients with limited knee functions appeared to fall more frequently within the first three weeks postoperatively. Therefore, appropriate measures should be taken to protect the postoperative outcome. Physical therapy and patient behavioural training should be practiced perioperatively in patients at risk.

## 1. Introduction

Injuries to the lower extremities are common injuries that occur during sporting activities or in everyday life. According to Bednarski et al., the knee injury rate is highest in the 11–20 age group with equal sex distribution [1]. Meniscal injuries and injuries to the anterior cruciate ligament (ACL) account for the largest share [1,2,3,4]. After reconstruction of the aforementioned anatomical structures, the use of postoperative orthosis with a limitation of range of motion (ROM) under 90° of flexion as well as partial weight bearing is common [5,6].

Several studies observed falling after surgery of the knee. Most of these studies focus on total replacement of the knee. In a review, Di Laura Frattura et al. stated fall rates between 12 and 38 percent in the first 6 months after surgery [7]. The overall fall rate of patients without pre-existing illnesses was calculated to be about 20 percent [8]. Overall, there are only a few studies on the risk factors for a tendency to fall after surgical interventions [9,10,11,12]. An increased tendency to slip, trip or fall after surgical intervention has been seen in elderly patients [9,10,11,12]. To date, no difference in the postoperative fall frequency has been found in relation to the body mass index (BMI) or the American Society of Anesthesiology (ASA) score [11]. In older patients, the preoperative tendency to fall was also found to be a risk factor for postoperative falls [12]. The risk factors ‘fear of falling’ and musculoskeletal comorbidities showed a tendency towards significance [12].

Although it is widely used, there is no evidence for a protective factor of an orthosis in the literature. Several studies did not find a significant difference between patients wearing an orthosis and patients not wearing one [13,14,15]. The systematic review of Lowe et al. states the positive benefit on the in vivo kinematic [16].

The aim of the present study was the prospective recording of falls and the calculation of fall rates after arthroscopic interventions, in general during the immobilization phase, and the evaluation of possible risk factors as well as the recording of fall-related re-interventions.

## 2. Materials and Methods

This prospective study was conducted from February 2019 to March 2021 at a maximum care hospital with a department for sports orthopaedics and arthroscopic surgery. During the entire study period, 200 patients underwent arthroscopic knee repair. 

The inclusion criteria of the study were a minimum age of 18 years and a performed arthroscopic reconstructive knee surgery—regardless of the anatomical structure—, which necessitated partial weight bearing of 15 kg and the usage of a rigid orthosis with a range-of-motion (ROM) limitation of full extension with restricted flexion. Exclusion criteria of this study were a lack of consent or ability to consent to the study, rejection of the after-treatment plan and the orthosis, use of orthostasis-promoting medication, injuries to the opposite side, wound infection, thrombosis, neurological diseases, psychiatric diseases and diabetes mellitus. 

At inclusion into the study, the age, gender, height and BMI as well as data on injury and therapy were recorded. Postoperative pain was measured by using the NRS (numeric pain rating scale). The pain was rated from 0–10 with 10 as maximum pain and 0 without pain [17]. A score median has been calculated and used for analysis. Moreover, the ROM (range of motion) was checked every week and was analysed 6 weeks postoperatively with a comparison of the maximum range of flexion and extension. Every week, the difference of knee swelling was measured using the circumference 10 cm above the patella in comparison with the non-operated knee. The median was calculated for every participant to compare it in the groups of falls and no falls. The duration of wearing the orthosis was recorded in hours every week, and the median was compared statistically.

The patients were assessed with different assessment tools at different times of the study. Preoperatively, IKDC (International Knee Documentation Committee), the fall risk assessment STRATIFY, the XSMFA-D (Short Musculoskeletal Function Assessment) and the Tegner Activity Scale were completed. [18,19,20,21] Furthermore, IKDC was checked every week. All these scores were secondary scores, as the primary aim of the study was to evaluate the fall rates and the tendency to fall. The IKDC is a tool to measure the function of the knee and has a score range of 0–100%. A total of 100% is a normal function of the knee, and a low score is a marker for an impaired function of the knee [18]. The STRATIFY score is divided into five questions. Every question aims to find a risk of falling. The maximum score is five points, zero points implies no falling tendency, one point is a moderate falling tendency, and two points or more represent a high tendency to fall [19]. The XSMFA-D consists of many single-choice questions, which are converted into numbers. The overall score ranges from 0–100, and a higher score indicates a worse function of the knee joint [20]. The Tegner score is a score to evaluate the activity of every participant. The score ranges from 0–10, where zero is doing no activities at all and 10 is performing professional competitive sport [21].

After surgery—regardless of which anatomical reconstruction was in question—, each participant received a four-point-rigid-knee-orthosis M.4s comfort (Fa. Medi, Bayreuth) with ROM limitations adapted to the operation: meniscal suture; ACL reconstruction in combination with meniscal suture or medial patellofemoral ligament (MPFL) reconstruction—extension/flexion 0-0-60° for 3 weeks and 0-0-90° for another 3 weeks; ACL reconstruction—0-0-90° for 6 weeks. In addition, all patients were mobilized postoperatively with physiotherapeutic support on two crutches, applying a partial load of 15 kg. The physiotherapists could start lymph drainage on the second postoperative day. In the case of meniscal sutures, passive movement was allowed for the first 3 weeks, and active movement with weight bearing was allowed after 6 weeks. In the case of ACL reconstruction or MPFL reconstruction, active movement was allowed from the second postoperative day. All patients received manual lymphatic drainage, local cryotherapy, as well as range-of-motion physiotherapy during the follow-up-period. 

Over a period of six weeks after discharge, the frequency of falls or near-falls (defined as losing balance and applying one’s full weight to prevent a fall), the range of motion, the NRS and the length of time during which the orthosis was worn were queried in a weekly telephone interview. Possible complications after falls were evaluated immediately by means of a clinical examination on the day after the fall, sonography, X-ray and MRI. 

For statistical analysis, falls and near-falls were combined, as both have an elevated risk of injury.

A post-hoc power analysis was conducted with a power level of 0.8 as well as a significance level of 0.05 for significant data, which has been fulfilled for XSMFA-D scores in the fall and no fall groups (sample size of 52).

The data were obtained prospectively and analysed retrospectively. The statistical analysis was performed using SPSS Statistics for Windows (Version 25; IBM Corp, Nuremberg, Germany). All *p*-values described are tested paired and are rated as statistically significant by *p* < 0.05. Data are presented as a mean ± standard deviation and range unless otherwise specified. Normal distribution was tested using the Shapiro–Wilk test. To evaluate statistically significant differences, the t-test for independent variables, the Mann–Whitney–U-test or the chi-square test was used, depending on the scale level and the extent of the normal distribution. 

## 3. Results

Of the 200 operated patients, 60 could be included in the study. A total of 9 study participants were lost during the follow-up, and 51 patients (39 male, 12 female) could be assessed in the analysis. The participants had a mean age of 31 years (19 years–57 years). The average BMI was 25.1 (16.6–37.2). There were no differences in the number of dropouts between the two study groups, fall event and no fall event.

As mentioned above, different types of surgery were included in the study: ACL-reconstructions (43.1%), meniscal sutures (27.5%), a combination of both of the aforementioned (13.7%), MPFL reconstructions (5.9%) as well as femoral cartilage repair (9.8%).

In this study, 39.2% (20 patients) of patients had a fall event and 60.8% (31 patients) did not. There were no significant differences in fall events between men and women. 

The majority (78.3%) of the falls observed occurred in the first three postoperative weeks. The fall rate was highest in the first postoperative week (30.4%) (Figure 1). Only 21.8% of all falls in the observed period occurred in the last three weeks (Figure 1).

The preoperative XSMFA-D scores showed a significant difference (*p* = 0.02) between patients with and without a fall event. The mean for the group with a fall event was 52.1 points, and the mean for the group without a fall was 45.3 points (Table 1). The other risk factors, such as BMI, age, gender, knee swelling, range of motion, NRS, wearing time, IKDC and Tegner score, showed no significant difference between the two groups (Table 1).

The evaluation of risk factors showed a significant difference between the fall and no fall group after the sixth week for the risk factors ROM, NRS and the wearing time of the orthosis. During the entire observation period, none of the patients required revision or intervention due to the fall event.

There was no difference between the kind of surgery in relation to the tested scores and risk factors, nor were there any differences between the surgery types and the frequency of falls.

## 4. Discussion

The most important finding in this prospective study is the high rate of falls (39.2%) after arthroscopic knee joint surgery within the first six postoperative weeks. In the first three weeks in particular, patients were at risk of falling. The overall condition of the musculoskeletal system as well as pain and the length of time during which the orthosis was worn were also identified as potential risk factors. Despite the high number of falls, no revisions or interventions had to be carried out.

Postural stability is an objective measurable parameter for the study of functional limitation. There are studies examining both postural stability after knee joint surgery and the relationship between postural stability and orthoses [22,23]. In the study by Gokalp et al., patients undergoing arthroscopic ACL reconstruction were assessed for fall risk using a posturographic analysis. One result was that patients with a preoperative ACL rupture are, on average, in the high-risk group for a fall. The highest risk of falling was measured in the study during the fourth postoperative week, where, on average, the patients could also be assigned to the high-risk group postoperatively. The result corresponds to our observations that the majority of falls (78.3%) occur in the first three weeks postoperatively (Figure 1). Since Gokalp et al. only collected data in the fourth, eighth and twelfth postoperative week, it might be possible to assess whether the actual peak in fall risk is found within the first three weeks [22]. One can postulate a higher risk of falls in the early postoperative phase, even after minimal invasive operations. 

In the present study, one can note that wearing a knee joint orthosis tends to have a positive influence on the number of falls. There is conflicting information about the effectiveness of a knee joint orthosis after arthroscopic knee surgery. Almost all studies relate to rehabilitation after cruciate ligament surgery, while no data is available for a broader spectrum. In their study, Derouin et al. describe the muscular contribution to rotational stability in ACL injuries and after ACL reconstruction [24]. The result of the study describes a 24% higher rotational stability of the knee joint while it is in a splint. The authors have concluded that an increased rotational stability of the orthosis group is triggered by increased muscle activity. It might therefore be assumed that an orthosis promotes muscular activity and thus minimizes falls. It must be noted, however, that there are several studies that do not find any significant differences in various parameters [13,16,25]. Bordes et al. examine athletes after ACL reconstruction and describe that there are no significant differences between the individual groups [13]. There is no significant difference in clinical outcome and no significant influence of the orthosis on the short-term complication rate. A systematic review by Lowe et al. shows that there could be potential benefits of orthoses for in vivo kinematics, such as protection of the joint implant [16]. 

This study on falls after minimally invasive arthroscopic knee surgery further confirms that patients with an impaired knee function have an increased risk of falling, after surgery on the knee joint, within the first three weeks, even when crutches and an orthosis are used. Therefore, after the prescription of appropriate measures such as an orthosis and crutches, direct attention should be paid to adequate training in handling the aids or mobilizing the patient under the guidance of a therapist. 

Further, it seems particularly important to pass on this information regarding the risk of falling to patients and to educate them accordingly. This study showed that the XSMFA-D could be a suitable questionnaire for identifying young patients at risk of falling after knee joint surgery. The introduction of a specific “fall training” could therefore also be particularly important for patients in the early rehabilitation phase. Consistent wearing of an orthosis and sufficient postoperative analgesia could have a positive effect on the fall rate. For this reason, possible problems with pain or the orthosis should be addressed intensively during follow-up treatment, especially in the first weeks. The exact influence of wearing an orthosis on the occurrence or prevention of recorded falls was not examined in this study. It can only be stated that no patients suffered injuries requiring revision or intervention during the fall with the orthosis. In this regard, studies with a larger number of cases and possibly a different study design should be developed. 

A strength of this study is that the rate of falls was examined prospectively through close controls. The weekly telephone interviews and surveys provided differentiated data, based on which the fall event rates in the immobilization period could be evaluated. In addition, this study focuses particularly on risk factors for slips, trips and falls. The fall event rate is explicitly determined and not just, as in many other studies, the mere probability of a fall event. In addition to the telephone surveys, close controls also included regular clinical examinations, another strength of this study. 

The limitations of this study are both the missing control group and the heterogeneity of the operations. Due to a missing control group, the orthosis cannot adequately be analysed as a protective factor. The heterogeneity of the different operations is a limitation that could affect falls differently. Nevertheless, in our opinion, the most important factor, which was also an inclusion criterion, for a fall was partial weight bearing as well as ROM-limitation after arthroscopic surgery. In addition, there was no further fall analysis. The place or the reason for falls was not analysed in this study and should be analysed in further studies. The occurrence of a recall bias is favoured by the telephone survey with regard to the fall events and the wearing time of the orthosis. Further limitations are possible comorbidities that have not been adequately considered and recorded.

## 5. Conclusions

There is a considerable fall rate during the immobilization period in patients with arthroscopic interventions. Risk factors seem to be the low condition of the musculoskeletal system as well as pain, while the role of wearing an orthosis in this context remains controversial. Intensified physical therapy and patient behavioural training should be recommended perioperatively to patients at risk.

## Figures and Tables

**Figure 1 jpm-12-01912-f001:**
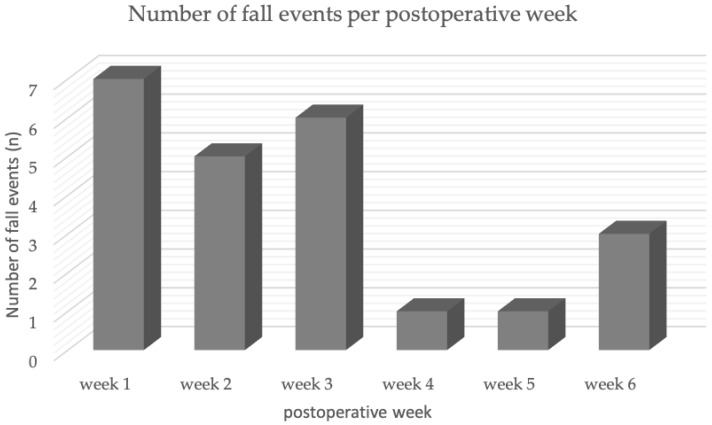
Number of fall events compared for each of the six weeks.

**Table 1 jpm-12-01912-t001:** Analyses of risk factors for postoperative fall events after arthroscopic reconstructive knee surgery.

	Total	Fall Event	No Fall	*p*-Value
Age (median ± SD, range)	31.2 ± 10.6, 19–57	31.1 ± 11.7, 19–57	31.4 ± 8.7, 20–52	0.5
BMI ^1^ (median ± SD, range)	25.1 ± 3.8, 16.6–37.2	24.7 ± 3.1, 16.6–30.4	25.7 ± 4.6, 20.8–37.2	0.4
NRS ^2^ (median ± SD, range)	3.2 ± 2.2, 0–8	3 ± 2.2, 0–8	3.4 ± 2.4, 1–8	0.6
Circumference ^3^ (median ± SD, range)	1.9 ± 1.4, −4–6	1.8 ± 1.3, −3.5–5	2 ± 1.5, −4–6	0.7
ROM ^4^ (median ± SD, range)	77.6 ± 19.4, 0–90	80.9 ± 15.9, 30–90	72.5 ± 23.5, 0–90	0.2
Wearing duration (median ± SD, range)	14 ± 6.5, 1–24	14.2 ± 6.5, 1–24	13.8 ± 6.8, 3–24	0.9
IKDC ^5^ (median ± SD, range)	45.7 ± 15.7, 23–79,3	49 ± 17.1, 23–79.3	40.6 ± 12.3, 23–65,5	0.1
STRATIFY (median ± SD, range)	0.7 ± 0.7, 0–2	0.7 ± 0.7, 0–2	0.8 ± 0.7, 0–2	0.7
XSMFA-D ^6^ (median ± SD, range)	47.9 ± 10.5, 22–71	45.3 ± 10, 22–68	52.1 ± 10, 31–71	0.02
Tegner (median ± SD, range)	5.6 ±2.1; 1–9	5.5 ± 2.5, 1–9	5.8 ± 1.6, 3–9	0.95

^1^ BMI: body mass index; ^2^ NRS: numeric pain rating scale; ^3^ Circumference: difference of circumference 10 cm above patella pole from operated and not operated knee; ^4^ ROM: range of motion; ^5^ IKDC: International Knee Documentation Committee; ^6^ XSMFA-D: Extra Short Musculoskeletal Function Assessment Questionnaire.

## Data Availability

Not applicable.

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
