# Peer review of "Risk and Influence Factors of Fall in Immobilization Period after Arthroscopic Interventions"

_jpm, 2022, doi:10.3390/jpm12111912_

Round 1
Reviewer 1 Report
The present manuscript entitled "Risk and influence factors of fall early after reconstructive knee joint surgeries" aims to evaluate the rate of falls in the immediate postoperative period corresponding to the maintenance of the knee orthosis, as well as to establish a relationship between the number of falls and various risk factors. This reviewer has the following comments and suggestions regarding it, which are detailed below:
TITLE
P1Ln2. The title does not correspond to the content of the paper. In the manuscript reference is made to “arthroscopic interventions” and not “reconstructive knee joint surgeries”. Likewise, the authors should consider the inclusion of “immobilization period” instead of “early after”.
ABSTRACT
P1Ln12. This statement regarding the possible protective effect of a rigid orthosis on falls is not developed in the introduction.
P1Ln14. Again, decide whether to refer to it as “reconstructive knee surgery” or as “arthroscopic interventions”. It must be homogeneous throughout the text.
P1Ln15. For a better understanding, detail the risk factors in parentheses.
P1Ln17. Present the data as percentages or as total numbers, but maintain homogeneity.
P1Ln22. What about the influence of the rigid orthosis as a protective factor against falls?
INTRODUCTION
P1Ln27. The introduction should be better developed. The authors should present the problem, defining the consequences of a fall in the postoperative period. Thus, the problem must be better specified and the risk factors described in a better grouped form (for example, type of injury, anthropometric characteristics, age, etc.). The authors go back and forth from one to another in this section without a clear common thread.
Likewise, since the study data is collected during the period of immobilization, the characteristics of this period and its relationship with falls should be written.
The abstract refers to the possible protective effect of the rigid orthosis on falls. It is not listed in the introduction.
P1Ln37. Missing reference.
P2Ln53. Include “rate” as detailed in the abstract.
MATERIALS AND METHODS
P2Ln59. These data correspond to the results section, where the subjects evaluated and finally included in the study are shown, apart from those who dropped out.
Was the sample size calculated based on any previous statistics?
P2Ln71. The study variables should be better detailed, grouping them as primary and secondary, as well as the inclusion of study covariates. The evaluation tools for each variable (scales or scores) must be explained. Likewise, the moments of evaluation of each variable must be made clearer. Was the place of the fall (home, street, etc.) or the reason for it considered?
P2Ln73-74. Maintain the order of variables in the text and in tables.
P2Ln76. Tegner or Tegener? It appears different here and in Table 1.
P2Ln77. For a better understanding of the paper, the authors should explain how the postoperative treatment was regarding the use of the orthosis and the physiotherapy treatment in a separate paragraph of the study variables.
P2Ln95. Provide reference for this statement.
RESULTS.
P3Ln107. Review the number of dropouts, and if it was significant depending on whether they belonged to the group of falls or not.
P3Table1. What is the meaning of this table? It could be irrelevant. Delete if so. Normally, what is presented is the baseline data between the groups (fall or no fall) to determine if they were homogeneous at the beginning of the study. On the other hand, BMI appears in the table legend, but not in the table itself.
P3Ln113. It would be important to present a table or figure showing the number of falls based on the postoperative week. This text information seems incomplete without that information.
P3Ln115. XSMFA-D is spoken of as the only differentiating risk factor between both groups. It is not clear from the title of this table 2 the moment of evaluation. To establish a relationship between the number of falls and these risk factors, it should be compared with the baseline values ​​at the beginning of the study, and not at the sixth week after the end of the study. In this line 116 it is said that the evaluation is at the sixth week, so it is methodologically wrong.
P4Ln132.135. In certain paragraphs of the manuscript it is difficult to clarify what is being explained, since there are no references to tables, details of statistical values, or simply an order in the presentation of the results based on the risk factors.
DISCUSSION
P4Ln143. Please compare this fall rate with the fall rate for knee replacement surgery, for example, to understand the magnitude of the results.
P5. In general, the discussion should be better developed. The authors continue to compare the data of the variables between the beginning and the end of the study. According to the objectives of the study, where the aim is to establish a relationship between falls and various risk factors, it is not pertinent to evaluate at six weeks, since a direct relationship cannot be established between this value and any fall in the first week, for example.
CONCLUSIONS
P6Ln227. Caution should be taken not to refer to "young patients", since in the inclusion criteria of the study it was only required to be of legal age.
P6Ln229. The authors establish inconclusive relationships with the results of the study.
Author Response
Please see the attachement

Reviewer 2 Report
Interesting paper deals with rarely investigated consequences of a fall after knee arthroscopy in younger people, as mostly older, rheumatic patients are treated. The literature review is comprehensive, the cited sources are relevant and up-to-date. Methods are well described, and limitations are listed. The theory and results support discussion and conclusions.
Author Response
Point: Interesting paper deals with rarely investigated consequences of a fall after knee arthroscopy in younger people, as mostly older, rheumatic patients are treated. The literature review is comprehensive, the cited sources are relevant and up-to-date. Methods are well described, and limitations are listed. The theory and results support discussion and conclusions.
Response: Thank you very much for your comment on our paper. We really appreciated it.
Round 2
Reviewer 1 Report
The manuscript has been carefully reviewed by the authors and considerably improved. It would have been positive to receive the manuscript without changes control and without comments in the margin, to be able to read the text in a better way. Even so, this reviewer suggests some last suggestions to better clarify some aspects, which are detailed below:
Point 6. With regard to this point, there is already a basic methodological problem, and that is that the authors detect that the objective of relating falls to this protective factor of orthoses cannot be analysed due to the lack of a control group. It should be reflected in the limitations of the study.
Point 14. It should be noted in M&M that the sample size was not calculated prior to the study, but that a post-hoc calculation subsequently determined that 52 subjects was the correct size.
Point 17. Please, it would be interesting for future research to leave this lack reflected in the limitations.
Point 22. Please note in the text that there were no differences in the number of dropouts between the two study groups.
Author Response
Response to Review report (Reviewer 1, round 2)
The manuscript has been carefully reviewed by the authors and considerably improved. It would have been positive to receive the manuscript without changes control and without comments in the margin, to be able to read the text in a better way. Even so, this reviewer suggests some last suggestions to better clarify some aspects, which are detailed below:
Thank you very much for the hint. The next review will be without the markups. This way it is more readable.
Point 6. With regard to this point, there is already a basic methodological problem, and that is that the authors detect that the objective of relating falls to this protective factor of orthoses cannot be analysed due to the lack of a control group. It should be reflected in the limitations of the study.
Response 6: Thank you for the constructive critics. We implemented it in the limitation section.
Point 14. It should be noted in M&M that the sample size was not calculated prior to the study, but that a post-hoc calculation subsequently determined that 52 subjects was the correct size.
Response 14: It is a good point, thank you. That’s why we implemented it in M&M and result scetion.
Point 17. Please, it would be interesting for future research to leave this lack reflected in the limitations.
Response 17: Thank you for the idea. As already stated, we didn’t check it. But we will check it in further studies. We also wrote it in the limitation section.
Point 22. Please note in the text that there were no differences in the number of dropouts between the two study groups.
Response 22: We wrote it in the M&M section. Thank you for all your great ideas and critics. The study got way better through your suggestions for improvement.